# A prognostic matrix gene expression signature defines functional glioblastoma phenotypes and niches
Monika Vishnoi [1,2], Zeynep Dereli[3], Zheng Yin[4], Elisabeth K. Kong[3,5], Meric Kinali[6], Kisan Thapa[6], Ozgun Babur[6], Kyuson Yun [7,8], Nourhan Abdelfattah [7,8], Xubin Li[3], Behnaz Bozorgui [3], Mary C. Farach-Carson[9,10], Robert C. Rostomily [1,2,11] ✉ & Anil Korkut [3] ✉

Interactions among tumor, immune, and vascular niches play major roles in glioblastoma (GBM) malignancy and treatment responses. The composition and heterogeneity of extracellular core matrix proteins (CMPs) that mediate such interactions are not well understood. Here, we present an analysis of the clinical relevance of CMP expression in GBM at bulk, single-cell, and spatial anatomical resolution. We show that CMP enrichment is associated with worse patient survival, specific driver oncogenic alterations, mesenchymal state, pro-tumor immune infiltration, and immune checkpoint expression. Matrisome expression is enriched in vascular and leading edge/infiltrative niches that are known to harbor glioma stem cells. Finally, we identify a 17-gene CMP signature, termed Matrisome 17 (M17), which is a stronger prognostic factor compared to MGMT promoter methylation status as well as canonical subtypes, and importantly, may predict responses to PD1 blockade. Patient stratification based on matrisome profiles can contribute to the selection and optimization of treatment strategies.

Glioblastoma (GBM) is an aggressive disease wherein meaningful improvements in survival have yet to be realized in a significant number of patients[1]. A high degree of intratumor heterogeneity, invasive growth and myriad mechanisms of treatment resistance limit the effectiveness of standard of care chemoradiation, targeted precision approaches and immunotherapy[2,3]. Central to the success of precision therapy is the integration of molecular and/or cellular data that accurately captures therapeutic targets and prognostic markers. The *MGMT* promoter methylation status, actionable driver mutations and immune checkpoint status have emerged as candidate biomarkers to inform chemotherapy, targeted therapy, and immunotherapy, respectively[4]. Despite their conceptual appeal, precision therapy and immunotherapy have yet to significantly improve durable outcomes[1,5–11]. The lack of progress with these approaches, and limited efficacy of standard genotoxic therapies, reflects, in part, insufficient understanding of factors that impact GBM malignancy and treatment responses.

The genomic landscape in GBM is well described with established correlations between clinical properties and transcriptomic GBM subtypes termed Mesenchymal, Classical, and Proneural[12–14]. However, the functional and clinical relevance as well as diversity of the GBM extracellular matrix (ECM) at the anatomic, tissue, and cellular levels are not as well established[15–17]. The ECM is organized as a large network of proteins, collectively termed the matrisome[18,19], comprised of core matrix proteins (CMPs, including glycoproteins, collagens, and proteoglycans) and matrix associated proteins (MAPs)[18,20]. CMPs provide a structural/functional scaffold while MAPs interact with and regulate the CMPs to promote cell adhesion and signaling and regulation of ECM composition and structure[19,20]. The matrisome proteins play critical roles in regulating normal and pathologic processes, including cancer malignancy[21,22]. Mechanistically, matrisome-tumor interactions mediated by CMPs contribute to cancer phenotypes through ligand-receptor interactions and structural mechanobiological effects that directly regulate tumor and stromal cell signaling or

[1]Department of Neurosurgery, Houston Methodist Research Institute, Houston, TX, USA. [2]Department of Neurosurgery, Weill Cornell Medicine, New York, NY, USA. [3]Department of Bioinformatics and Computational Biology, MD Anderson Cancer Center, Houston, TX, USA. [4]Department of Systems Medicine and Bioengineering, Houston Methodist Neal Cancer Center, Houston, TX, USA. [5]Department of Statistics, Rice University, Houston, TX, USA. [6]Computer Science, College of Science and Mathematics, University of Massachusetts Boston, Boston, MA, USA. [7]Department of Neurology, Houston Methodist Research Institute, Houston, TX, USA. [8]Department of Neurology, Weill Cornell Medicine, New York, NY, USA. [9]Department of Diagnostic and Biomedical Sciences, School of Dentistry, The University of Texas Health Science Center at Houston, Houston, TX, USA. [10]Departments of BioSciences and Bioengineering, Rice University, Houston, TX, USA. [11]Department of Neurosurgery, University of Washington School of Medicine, Seattle, WA, USA. ✉e-mail: rrostomily@houstonmethodist.org; akorkut@mdanderson.org

indirectly modulate the tumor microenvironment (TME)[23]. A better understanding of the CMP landscape in the GBM TME is expected to reveal new insights into diverse mechanisms of GBM malignancy, facilitate optimization of existing GBM therapies as well as development of new therapeutic strategies. Of relevance to precision oncology, matrisome-cancer cell interactions are likely to modulate drug responses that are usually predicted solely based on mutation profiles. A deeper understanding of the matrisome-GBM interactome is also expected to refine the performance of ex vivo organotypic and 3D pre-clinical GBM models to predict treatment responses[24–31]. A critical first step towards achieving these goals is to identify the components of the CMP molecular landscape that correlate with patient outcomes, and clinical and molecular features of GBM malignancy.

Here, we undertook a comprehensive analysis of the genes encoding core matrix proteins (CMPs) in GBM. Through analysis of CMP gene expression patterns in the TCGA GBM dataset, we first identified three groups of GBM with matrix-high (M-H), matrix-low type a (M-La) and matrix-low type b (M-Lb) gene expression profiles. Importantly, the M-H profile predicted worse clinical outcomes and was associated with oncogenic processes, including epithelial mesenchymal transition (EMT), a pro-tumor immune signature and oncogenic signaling relevant to GBM malignancy. Using the IvyGap GBM database, we detected spatial enrichment of CMP-encoding gene expression in the vascular and infiltrative regions, suggesting that the CMP profiles may comprise a "code" that defines specific functional GBM niches. Consistent with this, single cell RNA expression analysis revealed enrichment of CMPs primarily in pericytes and endothelial cells with moderate expression in glioma cells and sparse expression in immune cells. Through a LASSO-analysis of the matrisome gene expression, we identified a 17-gene matrisome signature (M17) that can stratify GBM patients into M-H, M-I (intermediate) and M-L groups. The M17 signature and the resulting refined patient stratification can predict patient survival better than the MGMT promoter methylation and canonical subtype-based stratifications. The M17 signature can also potentially predict response to anti-PD1 blockade in recurrent grade 4 gliomas that are predominantly GBM cases. Our CMP analysis and the resulting M17 signature provide a potentially widely applicable marker to characterize GBM disease progression, phenotypes, and potentially therapy response in preclinical and clinical research.

## Results

### A compendium of multi-modal molecular and clinical data from GBM patients

To identify the GBM matrisome signature, we established a compendium of genomic, transcriptomic, and phosphoproteomic datasets. We analyzed the core matrisome proteins (CMPs, $n = 274$) including glycoproteins ($n = 195$), collagens ($n = 44$), and proteoglycans ($n = 35$)[18,20,32] (Supplementary Table 1) derived from Matrisome 2.0 database[19,20]. To account for the inter-patient matrisome heterogeneity, we included 157 IDH wild type (WT) GBM samples from 151 patients with varying coverages for genomic, transcriptomic, proteomic, and clinical data from the TCGA GBM dataset[12] (Supplementary Table 2). We mapped the spatio-anatomical heterogeneity of CMP gene expression using the transcriptomic data from the IvyGap GBM database (245 samples across 7 anatomic regions in 34 tumors)[13]. The matrisome heterogeneity across GBM tumor niches was further analyzed using single cell transcriptomics data from 201,986 glioma, immune, and other stromal cells in 16 IDH WT GBM tumors from patients[33]. The proteomic outcomes of the transcriptomic signatures were determined using mass-spectroscopy data and RNA sequencing data available for IDH-wild type GBM tumors ($N = 86$) in the CPTAC data repository. For each data modality, we integrated the relevant clinical and phenotypic parameters to establish multi-faceted interactions between the ECM and key GBM phenotypes, including immune, vascularization, cell signaling, and differentiation (e.g., EMT) states. We studied the therapeutic implications of the matrisome enrichment in GBM using transcriptomic and survival data from a clinical trial of 28 patients with resectable recurrent grade 4 glioma and treated with anti-PD1 therapy[34]. The resulting multi-modal

data compendium enabled us to establish a robust matrisome signature that predicts patient survival, disease characteristics and therapy responses in GBM (Fig. 1A).

### Core matrix gene expression signatures predict GBM patient survival

To investigate whether the interpatient heterogeneity of the core matrisome defines clinically relevant subgroups of GBM, we analyzed the distribution of CMP-encoding gene expression levels in the TCGA GBM dataset[35]. Unsupervised hierarchical clustering analysis identified three major clusters of GBM based on distinct CMP-encoding gene expression levels (Fig. 1B), termed as (i) matrix-high (M-H, $n = 44$ cases, 28% of the cohort), (ii) matrix-low (M-L) which was further classified as matrix-low type a (M-La, $n = 58$ cases, 37% of the cohort) and matrix-low type b (M-Lb, $n = 55$ cases, 35% of the cohort) (Fig. 1B, Table 1). Compared to the M-H sub-group, overall CMP-encoding gene expression levels were lower in the two M-L subgroups, M-La and M-Lb. These CMP sub-groups correlated with differential progression-free survival (PFS) and overall survival (OS), with M-H having the worst OS and PFS [Median PFS (months): M-H = 4.3, M-La = 7.0, M-Lb = 6.7, log-rank $p = 1.76\text{e-}2$; Median OS (months): M-H = 11.0, M-La= 13.0, M-Lb=16.1, log-rank $p = 1.34\text{e-}2$] (Fig. 1C). When M-La and M-Lb cohorts were combined into a single matrix-low group (M-L) and compared to M-H, the significant differences in PFS and OS remained (log-rank $p = 6.01\text{e-}3$ and $1.13\text{e-}2$, respectively) (Supplementary Fig. 1). In a multivariate analysis accounting for gender, age, canonical GBM subtypes defined based on interpatient transcriptomic heterogeneity, MGMT methylation status and treatment (chemoradiation vs radiation alone); M-H versus M-La or M-Lb independently predicted shorter survivals (log-rank $p = 3.0\text{e-}3$ and $2.7\text{e-}13$, respectively; Supplementary Table 3). Consistent with their known prognostic importance, unmethylated MGMT, radiation only versus chemoradiation treatment, and increased age also demonstrated significant negative associations with survival (log-rank $p = 3.33\text{e-}15$, $6.77\text{e-}12$, and $1.49\text{e-}09$, respectively; Supplementary Table 3).

We asked whether CMP patient subgroups are associated with the canonical transcriptomic subtypes of GBM (Mesenchymal, Classical, Proneural, Neural)[36] (Table 1). The M-H subgroup is enriched in the Mesenchymal GBM (mesGBM) subtype (Fisher's Exact test, $p < 1\text{e-}04$). On the other hand, the M-Lb subgroup is associated with the Classical GBM subtype (Fisher's Exact test, $p = 2\text{e-}03$), while the M-La subgroup is enriched in the Proneural subtype (Fisher's Exact test, $p < 1\text{e-}04$). To test whether the matrisome status is associated with prognosis independent of the canonical subtypes, we compared the survival of M-H vs. M-L patients within the mesGBM subtype. We observed significantly poor survival outcomes of patients with mesGBM/ M-H tumors compared to mesGBM/ M-L tumors (log-rank $p = 4.91\text{e-}2$ for PFS and $1.52\text{e-}2$ for OS) (Fig. 1D). This observation suggests that the CMP defined intertumor heterogeneity and subgroups (M-H vs M-L) are clinically relevant and the impact of M-H on survival is not driven by its enrichment in canonical transcriptional subtypes. Of note, recent revisions de-emphasize the Neural subtype due to concerns that it reflects potential contamination from normal neural tissue[37]. To test whether the relevance of matrisome composition was confounded by the Neural subtype, we repeated the survival analysis with exclusion of Neural subtype samples. In the absence of Neural samples, we observed similar overall and progression free survival profiles suggesting the matrisome composition and survival association is not impacted by Neural subtype (Supplementary Fig. 1C). Together these data indicate that differential expression of CMP genes provides a robust and independent prognostic biomarker of clinical outcomes in IDH WT GBM patients.

### Matrisome signature expression is associated with expression of mesenchymal and immune markers

We next analyzed the molecular correlates of each CMP sub-group. The M-H and M-Lb patient subgroups were significantly enriched with NF1 mutations and EGFR copy number amplifications, respectively (Fig. 2A). Mutations associated with the Proneural subtype (e.g., PDGFRA amplification) were not

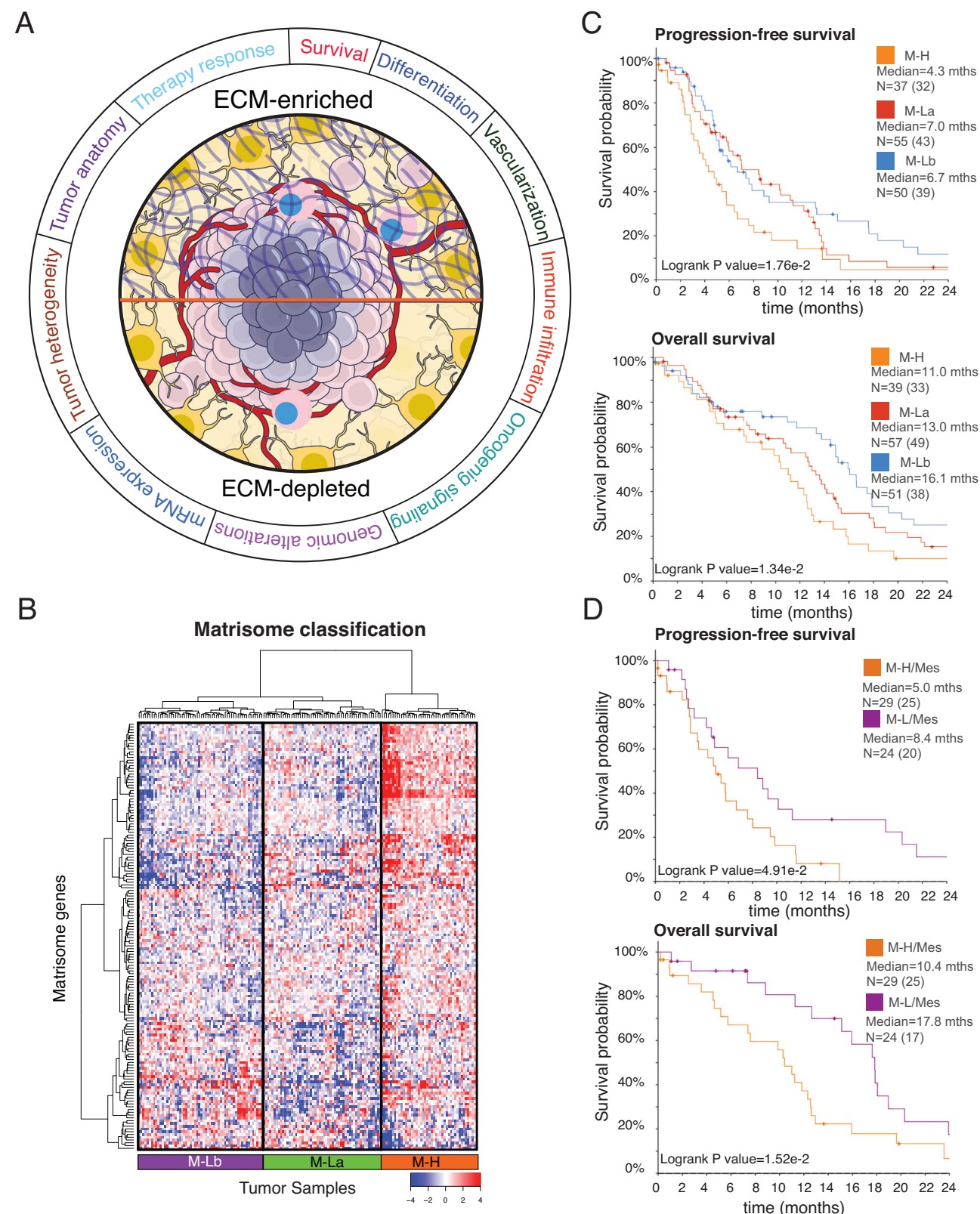

enriched in the M-La subgroup (Fig. 2A). Gene set enrichment analysis (GSEA) of differentially expressed genes (DEGs) in M-H versus the combined M-L sub-group identified the enrichment of transcripts associated with leukocyte migration, leukocyte activation, cell motility, angiogenesis, ECM organization, and cell adhesion in the M-H subgroup (Fig. 2B and Supplementary Fig. 2). The phospho-proteomics data (available for 23 M-H and 54

M-L samples for 191 total protein or phosphoprotein markers) enabled differential proteomics and pathway activation analyses (Fig. 2C and Supplementary Fig. 2B)[38,39]. PAI1 (Serpine 1), fibronectin (FN1) and caveolin1 are the most upregulated proteins in M-H while beta-catenin is significantly enriched in the M-L CMP subgroups. PAI1 promotes GBM invasion, is upregulated in mesenchymal GBM subtypes and is associated with shorter

**Fig. 1 | The landscape of ECM composition in GBM and patient survival. A** The association of extracellular matrix composition with key molecular and physiological states in GBM. **B** The hierarchical clustering of mRNA levels of core extracellular matrix genes (matrisome, N = 274) in 157 GBM tumors from 151 patients (source: TCGA, Level 3 mRNA sequencing data). The genes with low expression variation (standard deviation, σ < 1) across the tumors are filtered out. The final set included 154 genes that are most variant across samples. Three patient groups are identified as Matrisome high (M-H), Matrisome Low A (M-La) and Matrisome Low

B (M-Lb) based on overall matrisome expression. **C** The progression-free survival and overall survival of patients with M-H, M-La, and M-Lb GBM. N is the number of patients. **D** The progression-free and overall survival of patients with mesenchymal transcriptional subtype and M-H or M-L matrisome subtype of GBM. The M-L cohort includes both M-La and M-Lb. In all survival curves, the number of events (death or disease progression) is given in parentheses. The marks on survival curves represent censored data points. mths=months. The survival curves are truncated at 24 months (Supplementary Fig. 1B for complete curves).

**Table 1 | CMP subgroup correlations with demographic features of IDH wild-type GBMs in TCGA**

| | | M-H (44) | M-La (58) | M-Lb (55) | Total | *P*-value |
|---|---|---|---|---|---|---|
| Gender | Male | 28 | 42 | 31 | 101 | 0.084* |
| | Female | 16 | 15 | 24 | 55 | 0.117* |
| | Not available | 0 | 1 | 0 | 1 | NA |
| Age Median (Std dev) | Male | 59 ± 10.62 | 59 ± 13.71 | 63 ± 9.78 | | 0.294** |
| | Female | 63 ± 12.26 | 62 ± 15.62 | 60 ± 10.34 | | 0.743** |
| Karnofsky Performance Score | M-H = 29; M-La=42; M-Lb=46 | 80 ± 12.45 | 80 ± 14.13 | 80 ± 15.27 | | 1.000** |
| Vital status | Living | 11 | 19 | 20 | 50 | 0.473* |
| | Deceased | 33 | 38 | 35 | 106 | 0.259* |
| | Not available | 0 | 1 | 0 | 1 | NA |
| Gene Expression Subtype | Proneural | 3 | 21 | 5 | 29 | <1e-04* |
| | Neural | 4 | 14 | 9 | 27 | 0.085* |
| | Classical | 5 | 13 | 23 | 41 | 0.002* |
| | Mesenchymal | 30 | 8 | 17 | 55 | <1e-04* |
| | G-CIMP | 0 | 1 | 0 | 1 | NA |
| | Not available | 2 | 1 | 1 | 4 | NA |
| Treatment | Chemo-radiation | 26 | 39 | 34 | 99 | 0.494* |
| | Chemotherapy only | 0 | 0 | 1 | 1 | NA |
| | Radiation only | 15 | 18 | 19 | 52 | 0.859* |
| | Other (Unspecified + Not available) | 3 | 1 | 1 | 5 | NA |
| MGMT Status | Unmethylated | 20 | 25 | 28 | 73 | 0.503* |
| | Methylated | 11 | 17 | 20 | 48 | 0.278* |
| | Not available | 13 | 16 | 7 | 36 | 0.290* |

*Fisher's exact test (see methods). **1-way ANOVA test.

overall survival[40]. Increased FN1 and caveolin1 are also associated with increased GBM malignancy[41–44]. Notably, among other pathways, the receptor tyrosine kinase (RTK) and core reactive[45] pathways are significantly enriched in M-Lb and M-H, respectively (Fig. 2C and Supplementary Fig. 2B). The RTK pathway activity in the M-Lb samples is consistent with epidermal growth factor receptor (*EGFR*) amplification enrichment in this sub-group. Activation of the core reactive pathway, comprised of stromal proteins, claudin-7, E-cadherin, beta-catenin, RBM15 and caveolin1, was first discovered as associated with poor survival in breast cancers[45]. Finally, consistent with EMT hallmarks of cell motility, angiogenesis, ECM organization and cell adhesion; tumors within the M-H subgroup are enriched with the EMT markers based on both transcriptomics- and proteomics-based scoring of EMT-signature genes[46–48] (*P* < 1e-4) (Fig. 2D and Supplementary Fig. 2C). In conclusion, CMP defined patient subgroups are enriched in genomic alterations consistent with their associated transcriptional subtypes, and importantly, demonstrate robust correlations with signatures of mesenchymal state, immune activation, and tumor malignancy (e.g., motility, adhesion, angiogenesis).

### Expression of CMP-encoding genes is associated with a pro-tumor immune infiltration

Based on the observation that immune-related processes were enriched in the M-H subgroup, we analyzed the characteristics of the tumor-immune

microenvironment (TIME) across CMP patient subgroups. We first quantified the overall immune infiltration using a methylation-based immune infiltration score for each tumor sample[49] and compared it across CMP patient subgroups. We observed a significantly higher immune infiltration in the M-H subgroup compared to the M-L subgroup (Fig. 3A). We further assessed the enrichment of specific immune cell types within the TIME by analyzing the RNA-based CIBERSORT deconvolution results available for 157 GBM patient samples from the TCGA repository[50]. This analysis identified an enrichment of pro-tumor immune cell types, including M2 macrophages, neutrophils, resting NK cells and regulatory T cells (Tregs) in M-H tumors (Fig. 3A and Supplementary Fig. 3A). Analysis of 31 immune checkpoint (ICP) genes that we curated based on the FDA-approval status and clinical trials of immunotherapies[51] revealed that expression levels of genes encoding therapeutically actionable immune checkpoints also correlated with CMP subgroups and immune infiltration (Fig. 3B and Supplementary Fig. 3B). Out of these 31 ICP genes, mRNA levels of 17 genes are significantly higher in the M-H subgroup (Supplementary Fig. 3B). The difference in the expression of *CD276* (B7-H3), an orphan receptor with immune-suppressive, angiogenic and EMT functions, between M-H vs M-L subgroups is most significant (Supplementary Fig. 3B). By comparing CMP subgroup-specific expression patterns of known immune receptor-ligand pairs, we mapped putative functional interactions that could be relevant

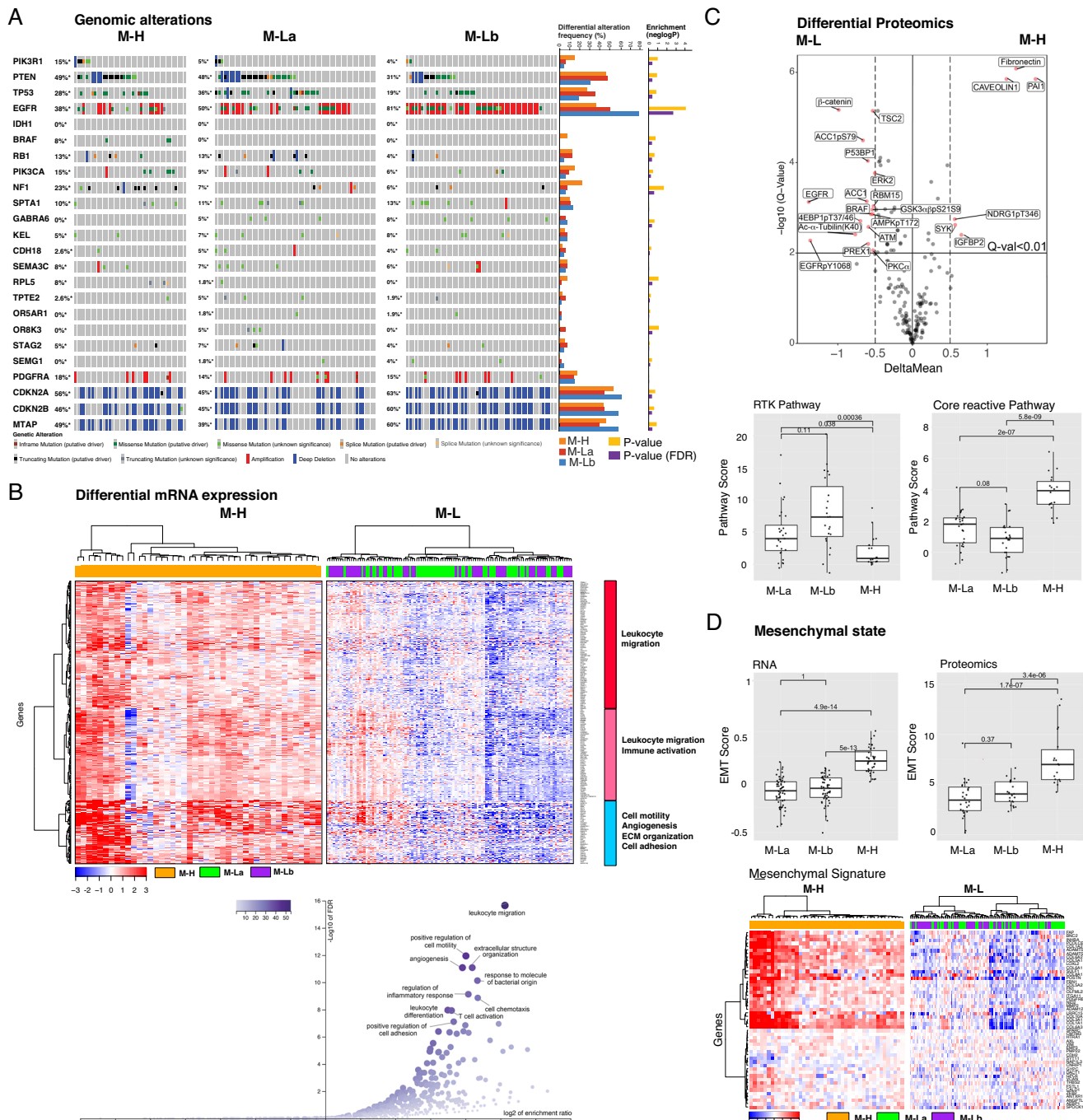

**Fig. 2 | The multi-omic landscape of matrisome-enrichment in GBM.**
**A** Mutational and copy-number alterations in GBM patients that are stratified by the matrisome subgroups. The *p*-values are based on two-sided Fisher's exact-test (FDR-adjusted using the Benjamini–Hochberg (BH) method) and quantify the significance of the alteration enrichment across the matrisome groups. **B** The differential analysis of mRNA expression between M-H vs. M-L groups. Genes whose mRNA levels are significantly higher (Bonferroni-adjusted $P < 0.001$ in a two-sided non-parametric Wilcoxon-test and log fold-change > 1.00) in the M-H group are included in the analysis. Three differentially expressed gene groups are defined based on a hierarchical clustering of mRNA expression levels in the M-H cohort. The functional enrichment within each group is annotated based on a gene set enrichment analysis (Web-Gestalt method). **C** The differential phosphoprotein and total protein levels in M-H vs. M-L tumors based on reverse-phase protein array (RPPA) profiling of the TCGA cohort ($n = 77$)[38,39]. On the volcano plot, the x-axis is the difference of mean expressions across the samples ($<X>_{M-H}- <X>_{M-L}$) and the *y*-axis is the *p*-values based on a two sided *t*-test (adjusted with Bonferroni correction) that quantify the significance of protein expression differences between M-H and M-L (M-La + M-Lb) groups. The boxplots quantify the differential pathway activities. The pathway activity scores are defined as cumulative expression levels of proteins that function in the corresponding pathways as defined in ref. 46. The activities of receptor tyrosine kinase (RTK) and core reactive (representing stroma gene expression) pathways are significantly different between the M-H vs. M-L groups. **D** The representation of epithelial vs. mesenchymal differentiation states in M-H vs. M-L groups. The proteomic[46] and transcriptomic[47] markers of EMT are used to calculate the EMT scores as defined in the respective references. The boxplots demonstrate the statistically significant differences between the M-H vs. M-L groups (*p*-values are based on a two-sided Wilcoxon-test). The heatmap visualizes the differential mRNA expression of mesenchymal genes in the matrisome groups (see Supplementary Fig. 2C for the epithelial gene expression signature).

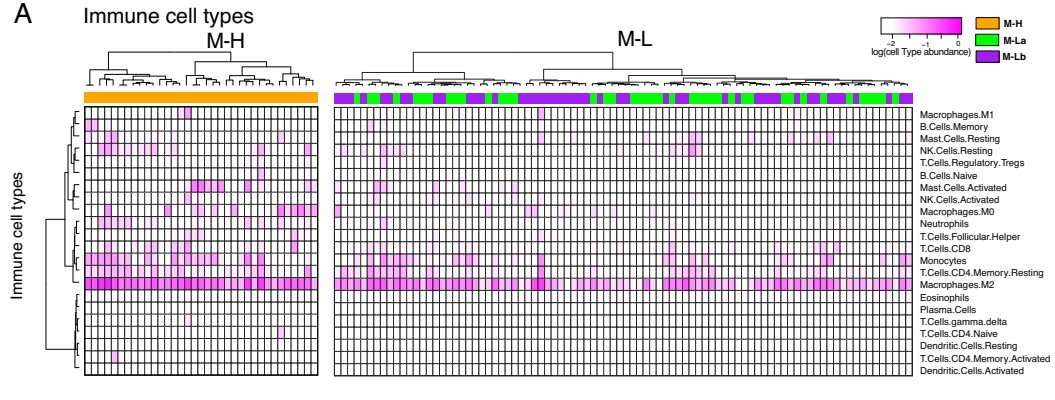

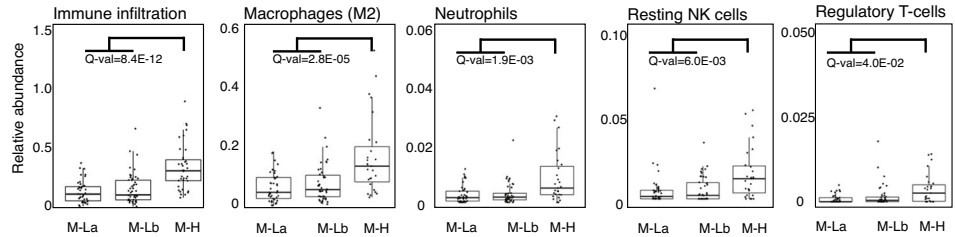

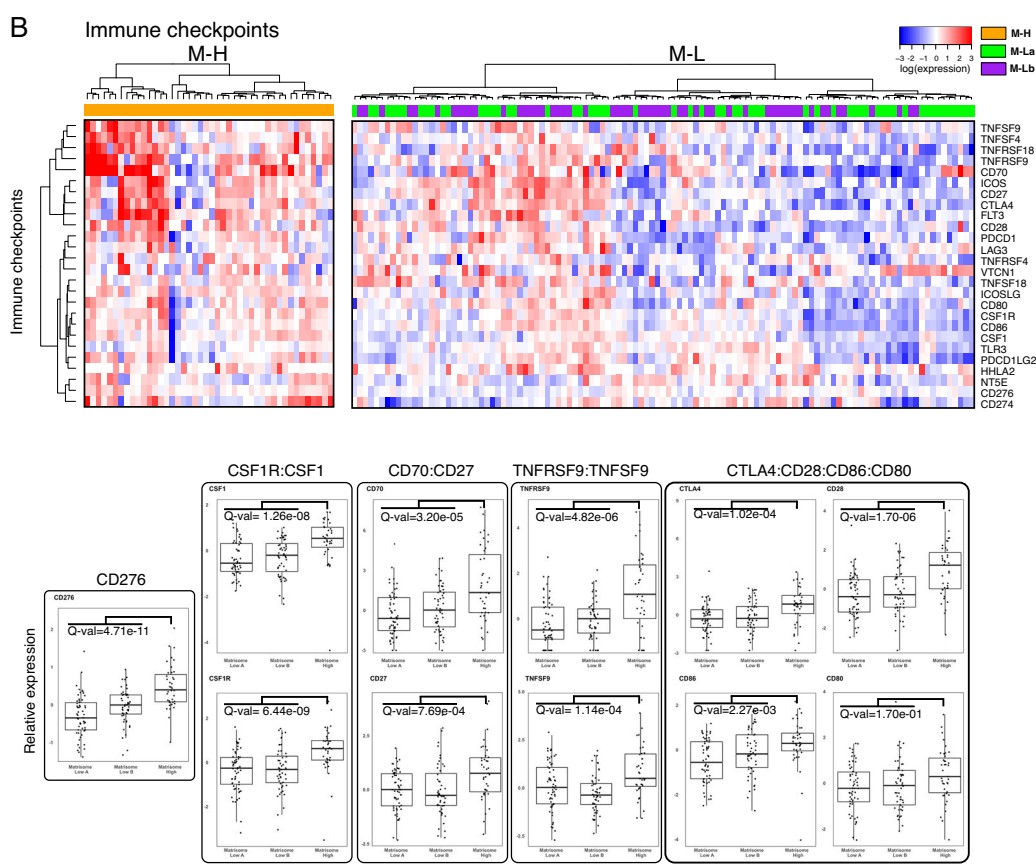

within the GBM tumor-immune interface[36]. In the M-H subgroup, we identified significant enrichment of CSF1R:CSF1, CD70:CD27, TNFRS9:TNFS9, CTLA4:CD86/CD80 and CD28:CD86/CD80 receptor-ligand pairs (Fig. 3B). The higher expression of the CSF1R:CSF1 pair in the M-H subgroup is consistent with the M2 macrophage enrichment in this subtype and nominates a therapeutically actionable immune checkpoint axis for future translational studies guided by matrisome-based stratification of GBM patients. In conclusion, these observations indicate that the characteristics of the M-H TIME support a unique set of cellular and molecular features consistent with an immune suppressive

**Fig. 3 | The GBM tumor-immune interactions across matrisome subtypes.**
**A** Enrichment of immune cell types in the microenvironment of M-H vs. M-L tumors. Immune cell type fractions were quantified through CIBERSORT analysis. The immune infiltration level in each sample is quantified based on the methylation-based immune fraction scores[50]. The enrichment of each immune cell type is quantified through the multiplication of the total immune infiltration score and fraction of the corresponding immune cell type. The statistical significance is quantified through the two-sided Wilcoxon-signed ranked test. The resulting p-values for M-H vs. M-La and M-H vs. M-Lb are merged using the Stouffer method

and corrected for multiple hypothesis testing across all immune cell types using the Bonferroni method. The immune cell type levels with significant differences across groups are shown in the box plots. **B** Distribution of immune checkpoint expression across M-H vs. M-L tumors. We analyzed the mRNA expression of immune checkpoint molecules for which targeting strategies are in clinical use or trials[51,89]. The statistical procedure is identical to that in (**A**). The immune checkpoint axes within which both the receptor and the ligand are statistically different between the groups are shown in the box plots except for the orphan receptor CD276.

microenvironment marked with increased pro-tumor M2 macrophages, Tregs, and neutrophils in M-H tumors.

### CMP-encoding genes are differentially expressed in vascular structures

To identify how CMP expression varies across different anatomical niches of GBM, we analyzed the localization and heterogeneity of the CMP-encoding gene expression patterns in GBM patients. The analysis of mRNA expression in 245 samples from 7 anatomical regions across 34 tumors in the IvyGap GBM dataset permitted spatial mapping of CMP gene expression to distinct anatomic/histologic domains, including leading edge (LE, $n = 16$), infiltrative tumor (IFT, $n = 21$), cellular tumor (CT, $n = 104$), microvascular proliferation (MVP, $n = 22$), hyperplastic blood vessels (HPBV, $n = 20$), pseudopalisading cells around necrosis (PAN, $n = 39$) and perinecrotic zone (PNZ, $n = 23$)[13]. Through a supervised hierarchical clustering analysis, we identified distinct gene expression clusters localized in vascular (MVP/HPBV) and infiltrative tumor-brain regions (IFT/LE) with a less distinct enrichment in regions associated with necrosis (PAN/PNZ) or solid cellular tumor (CT) without vascular or necrotic features (Fig. 4A). Although marked heterogeneity in the CMP-encoding gene expression is evident in samples from individual patients, the patterns of CMP-encoding gene expression within similar anatomic regions were largely conserved across GBM patients (Supplementary Fig. 4A). This finding indicates that intratumoral heterogeneity is primarily due to regional enrichment of specific sets of CMP genes in vascular, infiltrative, and solid cellular tumor regions.

We studied the specific GBM cell types underlying the CMP-encoding gene expression patterns using a large single-cell RNA sequencing dataset with cell type annotations[33] (Supplementary Table 4). Consistent with the localization in vascular regions, CMP-encoding gene expression was markedly increased in pericytes, which are present at the walls of capillaries, and mediate both vascularization and immune cell entry to central nervous system (CNS). Significant CMP gene expression was also observed in endothelial cells, while expression levels were modest in glioma cells and negligible in myeloid, B or T cell clusters (Fig. 4B). A unique gene expression cluster was identified in cells with markers of the myelinating cells of the CNS, oligodendrocytes (Fig. 4B); interestingly, in the IvyGap dataset, the genes in this cluster were highly expressed in the LE/IFT niche, moderately expressed in CT, but virtually non-expressed in MVP, HPBV, PAN and PNZ niches (Supplementary Fig. 4B). To further examine the relationship between CMP expression and tumor vascularity, we analyzed the expression of vascularization markers in GBM samples using a previously reported cancer vascularization signature[52]. The analysis identified enrichment of vascularization markers, suggesting higher degrees of vascularization, in the M-H vs M-La and M-Lb subgroups (Fig. 4C). In summary, the matrisome genes are predominantly expressed in vascular structures including endothelial cells as well as pericytes and, in turn, associated with increased vascularization in GBM tumors.

### A matrisome signature establishes a prognostic marker for GBM

Given the robust associations of the M-H subgroup with patient outcomes and tumor phenotypes, we sought to generate a minimal M-H specific gene set as a potential signature to guide assessment of GBM prognosis and treatment responses. Guided by a Lasso (Least Absolute Shrinkage and Selection Operator) regression that identified the most discriminating genes

between the matrisome subgroups, we selected 17 CMPs-encoding genes highly enriched in the M-H subgroup (Supplementary Fig. 5A–B). With hierarchical clustering analysis, we defined three distinct patient clusters across the TCGA GBM cohort ($N = 157$) distinguished by high, intermediate, and low expression of these 17 genes (Fig. 5A) and designated them as M17-H ($N = 32$), M17-I ($N = 55$) and M17-L ($N = 71$) subgroups, respectively. Importantly, these subgroups retained prognostic significance for OS and PFS, with M17-H having the worse outcomes (log-rank $p = 4.12$e-3 and 5.39e-4, respectively). Moreover, the inverse relationship between the signature expression and survival is monotonous across the subtypes such that PFS and OS for M17-H, M17-I and M17-L were 3.9, 5.3, and 8.4 months and 10.4, 12.5, and 14.9 months, respectively (Fig. 5B). Based on an assessment of Hazard Ratios (HR) obtained from the survival analysis of the same cohort, we observed that the matrisome gene signature provides a stronger prognostic factor compared to the MGMT methylation status, a well-established marker for GBM prognosis as well as the three canonical GBM molecular subtypes (Fig. 5C, Supplementary Fig. 5C–F), In summary, the 17-gene signature, which we referred to as the "matrisome 17 (M17) signature", provided a refined, potentially widely applicable prognostic GBM marker which is a stronger metric for assessment of disease progression compared to existing markers in GBM.

Next, we characterized the heterogeneity and proteogenomic landscape of the matrisome signature in GBM patients. We mapped the heterogeneity of the M17-H signature through a single-cell transcriptomics analysis (Fig. 5D, Supplementary Fig. 5G). Consistent with the enrichment of the expression of CMP-encoding genes in the vascular niche, each gene in the signature is expressed at high levels in pericytes and/or endothelial cells, with a few exceptions (e.g., *COL22A1*). Although at less substantial levels, there is also variable co-expression of these 17 genes in other cell types, most predominantly in glioma cells. We investigated the proteogenomic reflection of the M17-H signature using transcriptomic and mass spectroscopy-based proteomic data from matched samples of GBM patients (CPTAC GBM database, $N = 86$ after filtering out IDH-mutated and/or poor-quality samples based on pathology assessment)[53]. Similar to the analysis of the TCGA-cohort, we observed three clusters in both proteomic and transcriptomic analysis that corresponded to M17-H, M17-I, and M17-L subgroups (Supplementary Table 5). The 17 gene matrisome signature is highly associated with poor survival, particularly for patients whose tumors are in the consensus M17-H subgroup (i.e., highly expressed signature at both protein and RNA levels, see Methods) compared to the patients with M17-L tumors (median survival times, M17-H: 365 days; M17-L: 651 days, log-rank $p = 0.069$, Fig. 5E). Although the sample size was substantially smaller compared to the TCGA cohort, the analysis of the mRNA and the protein data individually also demonstrate strong trends of poor survival outcomes when the M17 gene signature is expressed (Supplementary Fig. 5H). The results from the proteogenomic and single cell transcriptomic analyses are consistent with those from TCGA- and IvyGap-based CMP gene-expression analyses for the patient subgrouping (M17-H, M17-I, M17-L), survival outcome trends, and anatomic distribution of the matrisome signature expression.

### Matrisome correlates with immunotherapy response in recurrent grade 4 gliomas

We asked whether the matrisome gene signature correlates with responses of glioma tumors to immunotherapy. We focused on a cohort of patients

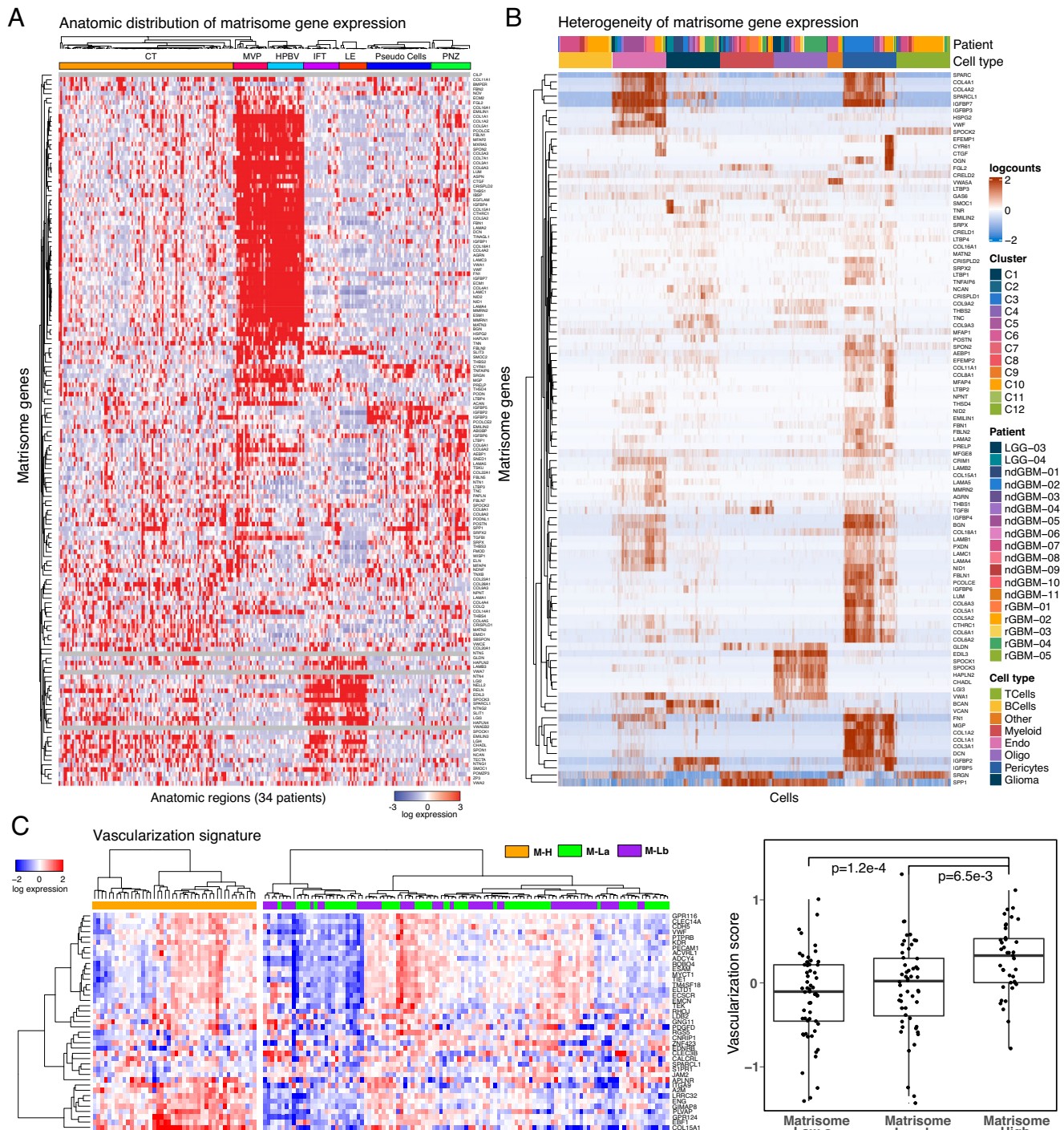

**Fig. 4 | The spatial compartmentalization and heterogeneity of matrisome expression. A** Supervised hierarchical clustering of anatomically resolved matrisome gene expression across 245 RNAseq samples in 34 patient tumors. **B** The heterogeneity of matrisome gene expression based on single cell RNA transcriptomics (GSE182109)[33]. **C** The heatmap representation of the vascularization gene signature expression in M-H vs. M-L tumors (left) and the quantitative comparison of the vascularization in M-H vs M-La and M-Lb based on the vascularization score as defined in ref. 52. The *p* values are based on a two-sided non-parametric Wilcoxon test.

with recurrent and surgically resectable recurrent grade 4 gliomas (N = 28, 24 IDH-wt GBM, 4 IDH-mutated grade 4 glioma), which were profiled for mRNA expression, and received neoadjuvant and/or adjuvant anti-PD1 therapy (Ivy Foundation Early Phase Clinical Trials Consortium)[34]. The survival time covered the period from trial registration prior to therapy and surgery to second progression or death, respectively for patients with resectable recurrent disease. The unsupervised clustering of mRNA levels of the 17 CMP gene signature partitioned the patients into M17-H (N = 19)

and M17-L (N = 10) groups (Fig. 5F). After anti-PD1 therapy, the patients with M17-H tumors had significantly shorter PFS and OS than patients with M17-L tumors (Fig. 5F, Supplementary Fig. 5I, log-rank *p* = 0.05 for both PFS and OS). The patients who received the neoadjuvant plus adjuvant treatment and the adjuvant-only treatment were similarly distributed between groups, 9 out of 19 patients with M17-H tumors and 5 out of 10 patients with M17-L tumors received the neoadjuvant plus adjuvant treatment (Supplementary Table 6). Survival analyses therefore were unlikely to

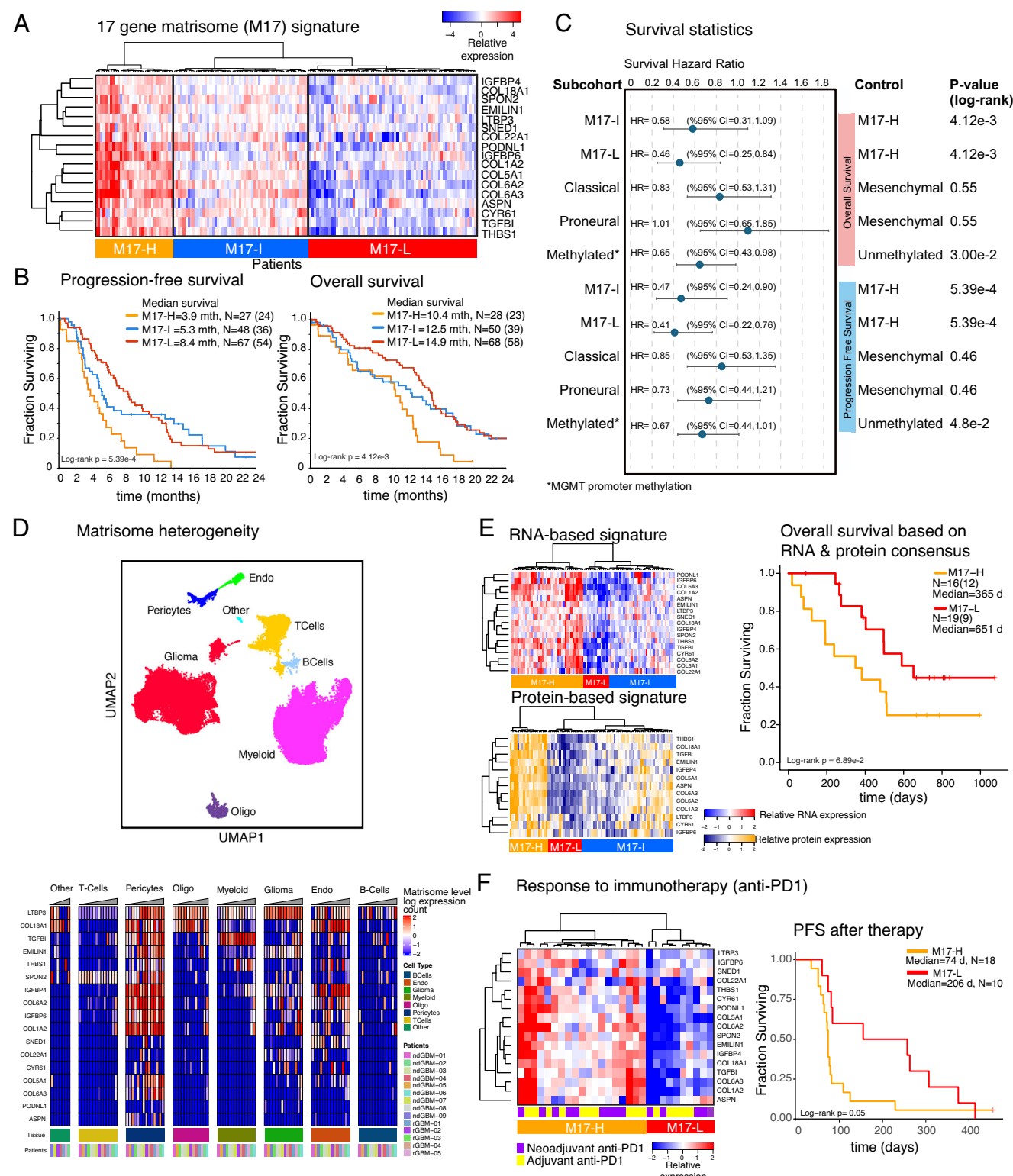

be confounded by the treatment difference between these two groups of patients. The survival analyses of patients stratified by PD1 and PDL1 expression did not result in significant survival differences suggesting immune checkpoint expression is not as strong as the matrisome composition in predicting survival outcomes in patients treated with immunotherapy. There was, however, a trend for better survival outcomes for anti-PD1 treated patients with high PD1 expression (log-rank $P_{PFS} = 0.22$, $P_{OS} = 0.18$) (Supplementary Fig. 5J). The correlation between the patient

survival, anti-PD1 treatment and matrisome status also justifies future clinical trials involving well-controlled study arms including anti-PD1 treated vs. untreated cohorts, and different matrisome states particularly to deconvolute the pure prognostic vs. therapy response predictions by the matrisome signature. Such future trials with larger patient populations may establish a CMP-based clinical biomarker that could guide selection of GBM patients who are likely to benefit from immunotherapy, while excluding those who are likely therapy-resistant.

**Fig. 5 | Predictive matrisome signature and response to immunotherapy. A** The heatmap representation of the 17-gene matrisome (M17) signature based on the LASSO analysis to capture the most discriminant genes across the matrisome subtypes. The M17 signature further refines the M-H cohort into M17-H and resolves the M-La and M-Lb into matrisome Intermediate (M17-I) and low (M17-L) cohorts. **B** The overall and progression free survival of patients with M17-H, M17-I, and M17-L GBM as defined in (**A**). (mths=months, truncated at 24 months). N is the number of patients. The numbers in parenthesis designate number of events (i.e., progression for PFS and death for OS). **C** The Hazard Ratio (HR) analysis for stratification of patients and survival outcomes based on matrisome signatures, canonical subtypes and MGMT promoter methylation status. The p-values are based on the log-rank test for survival. The HR values are derived from the log-rank tests for pair-wise comparisons of subcohorts and the control groups. The error bars on the HR plots represent the 95% confidence interval. **D** The single cell transcriptomic analysis of the M17 signature. The distribution of cell types in the GBM

ecosystem (top, adapted from ref. [33]) and the distribution of the matrisome signature expression across cell types (bottom) based on single cell transcriptomics. **E** Transcriptomic (top, left) and proteomic (bottom, left) profiles of the M17 signature in the matched tumor samples within the CPTAC repository. The proteomic analysis involves the 13 proteins that are encoded by genes within the M17 signature and are presented in the CPTAC mass spectroscopy data[53]. The comparison of overall survival between the M17-H vs. M17-L patient cohorts (right). The M17-H and M17-L groups in the survival analyses represent the patients whose signature expression levels are concordant across the protein and RNA profiles. (d=days) **F** The M17 signature levels correlate with responses to PD1 blockade. The heatmap representation of the mRNA expression in tumor samples from predominantly GBM, recurrent grade 4 glioma patients in the clinical trial with adjuvant and/or neoadjuvant anti-PD1 therapy[34] (left) and the PFS (from day of registration to progression per iRANO criteria or death) in M17-H vs M17-L patients that are stratified based on the M17 signature expression (d=days) (right).

## Discussion

In this study, we employed multi-modal molecular and clinical analyses of diverse, independent patient datasets (TCGA, IvyGap, CPTAC, and single-cell transcriptomics) to develop a new classification system for IDH wild-type GBM patients. Rather than canonical clinical and phenotypic subtypes, this new system is based on expression of core-matrisome protein (CMP) encoding genes in GBM patient tissues.

Here, we sought to establish the clinical and functional relevance specific to CMPs for GBM malignancy. Unsupervised hierarchical clustering analysis of TCGA RNAseq data identified subgroups stratified by high (M-H) or low CMP encoding gene expression (M-La and M-Lb or a combined M-L subgroup). We have made a series of critical observations that demonstrate that CMPs are highly relevant and essential for characterizing GBM stratification particularly for disease progression and survival. First, high CMP gene and protein expression (M-H) strongly correlated with shorter progression free and reduced overall survival across GBM patients. Second, while there is an enrichment of Mesenchymal subtype in the M-H, the Mes/M-H patients have significantly worse survival compared to Mes/M-L demonstrating the relevance of the CMPs independent of and within the canonical subtypes. Third, as shown in multivariate analysis, M-H independently predicted outcome when accounting for other known prognostic variables. Fourth, the M17 signature provides a manageable list of CMP genes which refines and optimizes the survival prediction power of the broad CMP-based classification. Most importantly, our M17 signature carries significantly stronger survival predictive power compared to well-established GBM markers such as *MGMT* promoter methylation and canonical subtypes. Moreover, the M17 CMP gene signature can stratify patients that may benefit more from immunotherapy as shown in a retrospective analysis of immunotherapy clinical trial data. We conclude these observations establish CMPs, particularly the M17 gene signature, as a highly useful and widely applicable biomarker with potential clinical applications for predicting GBM progression.

In addition to strong associations with disease progression, molecular and cellular phenotypes of the M-H CMP subgroup corresponded with mesenchymal features of GBM malignancy including EMT-enriched expression profiles, immune suppressive features, and increased vascularization. At the protein level, mesenchymal phenotypes of M-H were reflected in a significant increase in fibronectin, PAI1 and caveolin 1, activation of the core reactive signaling pathway and EMT score while at the transcriptomic level M-H exhibited increased EMT scores and mesenchymal gene signatures. The differential gene expression analysis identified enrichment of genes associated with leukocyte migration and immune activation in M-H versus M-L groups. This observation is consistent with the increased immune infiltration in the mesenchymal GBM subtype as well as the mechanistic links between EMT and immune suppressive tumor microenvironments in other cancers[54–56]. Concurrent enrichment of non-immune mesenchymal features of GBM including cell motility, cell adhesion, ECM organization and angiogenesis suggested potential functional

interactions may exist between upregulated M-H associated CMPs, the immune tumor microenvironment and vascular niches that collectively drive EMT/mesenchymal changes associated with GBM malignancy.

The immune tumor microenvironment of M-H GBMs featured increased overall immune cell infiltration, with specific increases in M2 macrophages, neutrophils, resting NK cells and Tregs[54,57,58]. M2 macrophages and Tregs are implicated in immune evasion in part through immune checkpoint-mediated signaling and suppression of CD8⁺ cytotoxic T lymphocytes[59]. The increase in resting versus activated NK cells may indicate a defect or reduction in innate immune responses[60]. The increase in tumor-associated neutrophils (TANs) in M-H is consistent with the observed increase in neutrophils reported in the mesenchymal GBM subtype[55]. Increased TANs generally predict poor patient outcomes, but their dual anti-tumor and immunosuppressive functions indicate that further study is required to determine the functional importance of increased TANs in the M-H GBM sub-group[61]. Increased peripheral neutrophil:lymphocyte ratios predict shorter GBM patient survival[62] and intratumoral neutrophils promote GBM malignancy in part through S100A4-mediated activation of glioblastoma stem cell (GSC) proliferation, invasion, and resistance to anti-VEGF therapy[33,63,64]. In addition to these cellular phenotypes, M-H expression data indicated that specific immune checkpoint genes and receptor-ligand pairs implicated in GBM malignancy and immune suppression are also increased in M-H GBMs, including CD276[65], CSF1R:CSF1[66], CD70:CD27[67,68], TNFRSF9:TNFSF9[69,70], CTLA4:CD80/86, and CD28:CD80/CD86[71] (Fig. 3B). Therefore, these observations of increased immune suppressive cell types and immune checkpoint expression support a potential role of M-H CMPs in promoting GBM immune evasion. Notably, concurrent increases in EMT/mesenchymal gene expression and immune suppressive phenotypes in the M-H group are consistent with correlations between mesenchymal GBM subtypes and immune suppressive TMEs[55,72]. Further, the recently recognized reciprocal cross talk between EMT and tumor immune landscapes identified in other cancer types[56,73] indicate a potential role for CMPs in orchestrating functional interactions between mesenchymal changes and the TIME.

Based on spatial analysis using the IvyGap dataset, CMP encoding gene expression was most strongly and uniformly enriched in vascular structures (MVP and HPBV) and to a less dramatic degree in the LE/IFT regions of the infiltrative tumor margins (Fig. 4A). These regions comprise distinct niches that harbor and maintain GSCs population[74–76], suggesting that CMP "code" may define specific anatomic and functional domains in GBM. Consistent with CMP gene enrichment in the vascular niche, single cell transcriptomic analysis identified pericytes and endothelial cells as primary contributors of CMP expression with modest contributions from glioma cells and oligodendrocyte cells (Fig. 4B). Analyzed across multiple tissue types, the THBS1, IGFBP4, CYR61 and Emilin1; components of the M17 gene signature, are reported as all highly transcribed in vascular tissues[77]. The enrichment of a prognostic ECM-signature expression in pericyte niches within recurrent GBM tumors has also been reported in a recent study[78].

**Fig. 6 | CMP crosstalk in the GBM vascular niche.** This model depicts CMP interactions and cellular crosstalk within the vascular niche that may promote oncogenic signaling and malignant GBM phenotypes. The model is supported by current findings that the high CMP expression in the M-H subgroup is tightly linked with EMT/mesenchymal change, immune suppressive phenotypes, enrichment within the vascular niche and vascular cells (endothelial and pericytes) and resistance to immune checkpoint blockade. The marked anatomic enrichment of CMPs in the vascular niche which harbors treatment resistant GBM stem cells, suggests that CMPs may promote clinically relevant features of GBM malignancy in part through trophic interactions with GSCs.

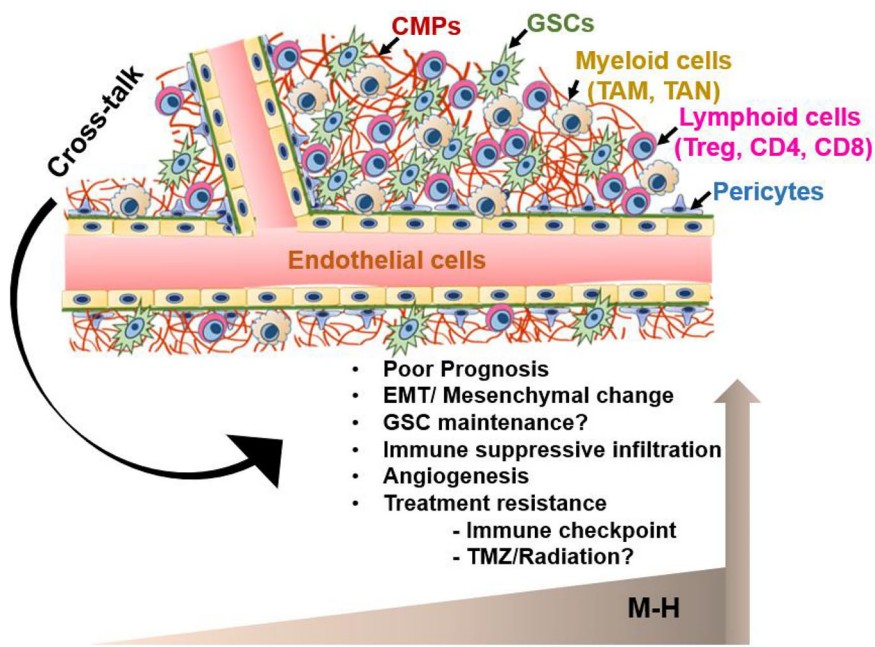

Among immune cells, CMP-encoding gene expression is negligible in B and T cells and is slightly higher but still sparsely existing in myeloid cells (Fig. 4B). The set of CMP genes localized specifically to the LE/IFT regions are uniquely enriched in oligodendrocyte-like cells (Supplementary Fig. 4), suggesting potential roles for these CMPs in promoting GBM invasion. Consistent with M-H CMP localization in vascular structures and the invasive front, mesenchymal GBM tumors are characterized by increased angiogenesis/vascularity[79] while invasiveness is a hallmark mesenchymal property of all cancers[80]. The potential interactions between the CMPs in these niches and GSCs may have clinical relevance through enhancing GSC growth, survival, and treatment resistance.

In other solid cancers, CMPs regulate angiogenesis and immune infiltration in the TME and modulate therapeutic responses to genotoxic and targeted drugs and immunotherapy[23,81–83], Through the current analysis of anatomical and cell-type specific CMP encoding gene expression, we provide new insights into the associations between GBM phenotypes, angiogenesis, and immune infiltrates. The strong correlations of CMPs with key oncogenic processes suggest that CMPs may function as a structural and functional hub critical to the integration of oncogenic crosstalk in the vascular and infiltrative niches. This crosstalk may promote malignant properties within the TME that also support GSC survival and treatment resistance, a hypothesis further supported by previous GBM studies[58,79,84,85]. A model of these putative interactions and their biological and clinical relevance is demonstrated in Fig. 6. Therefore, future functional testing of CMPs identified here may reveal therapeutic vulnerabilities in GBM that can be targeted with novel precision oncology paradigms. Given the central role of GSCs in GBM treatment resistance, progression, and dissemination, this new understanding of CMP composition specific to GSC niches provides a robust starting point for future investigation of potentially actionable novel mechanisms that modulate GBM malignancy and treatment responses. Further, a novel 17 gene M17-H CMP signature which reproduced the prognostic and phenotypic performance of the full CMP gene set. The M17 signature potentially predicted responses to immune checkpoint therapy in a small cohort of recurrent grade 4 glioma patients who are predominantly IDH-wt GBM. The results of this retrospective study warrants further validation of the M17 signature as a biomarker to predict treatment responses for standard chemo-radiation as well as immune therapy in well-controlled clinical trials particularly to deconvolute prognostic vs therapeutic predictive power. While no significant association was observed for PD1/PDL1 expression and survival outcomes, the trend for

PD1-expression and better survival outcomes in patients treated with anti-PD1 warrants future clinical studies in larger patient cohorts to investigate a combined matrisome and PD1 expression signature to reach better predictive power for therapy response.

Future comprehensive follow-up studies with multiplexing spatial gene and protein expression data at the single cell level can address challenges in this study inherent in deconvolution of cellular interactions from bulk tumor tissue and non-spatial single cell data. Larger data sets in varying clinical treatment contexts can help further define the biological and clinical relevance of the M-La and M-Lb CMP subgroups and identify CMP signatures that are enriched in other functionally important GBM domains such as the necrotic/peri-necrotic and solid tumor regions. Additional multi-institutional, large-cohort patient studies on GBM tumor samples are planned to determine whether inclusion of the matrix associated protein (MAPs) components of matrisome may provide unique or complementary prognostic or predictive information alone or in combination with the CMPs. Establishing the mechanistic relevance and functional impact of specific CMPs or CMP combinations in the context of complex interactions within GBM/GSC malignant niches is challenging as it requires multiplexing spatial transcriptomics and protein profiling from large set of clinically annotated archival or prospective tumor samples. Meeting this challenge and identification of actionable CMP driven targets may ultimately require development of 3D biomimetic models that incorporate CMPs within relevant GBM/GSC niches to test CMP effects on GBM signaling, phenotypes and therapeutic responses. To provide statistical power required to validate the M17 prognostic power and its predictive value for responses to specific therapies would require its prospective application in patients enrolled through a multi-center consortium of patients undergoing standardized treatments or as part of a multi-center therapeutic trial. These efforts will require development of a standardized prognostic assay that can be translated in the clinic. We have already demonstrated that the M17 signature is valid in both transcriptomic and proteomic domain through analyses of the CPTAC mass spectrometry data. This increases the confidence that the M17 signature can be translated into clinical applications. However, given the challenges associated with use of RNA sequencing and mass spectroscopy-based proteomics in clinical applications, there is a need for development of IHC based assay that can be implemented in the clinical setting. The validation of IHC assays representing the M17 (or a representative subset of M17) can lower the barriers for clinical implementation of CMP monitoring for patient prognosis and possibly drug response

evaluation in GBM. In conclusion, deeper functional and mechanistic characterization of the M17 gene signature will reveal its potential as a translationally relevant tool for predicting prognosis and response to chemotherapy or immunotherapy in GBM patients.

## Methods

### TCGA datasets-based multi-omics analyses

The TCGA datasets-based genomic, transcriptomic, and proteomic analyses were performed using data available from cBioportal (https://cbioportal-datahub.s3.amazonaws.com/gbm_tcga_pub2013.tar.gz, https://cbioportal-datahub.s3.amazonaws.com/gbm_tcga.tar.gz)[35,49,86]. As per 5th edition of WHO classification of CNS tumors, the absence of IDH mutations (IDH wild-type) defines grade IV Glioblastoma due to its validated and robust associations with distinct biological and clinical features of GBM malignancy compared with IDH mutant gliomas[87]. Therefore, tumors with IDH1 mutations were excluded from the TCGA analyses. The transcriptomic analysis included RNA expression data from 157 tumor samples of 151 GBM patients. We focused our study on the 274 core-matrix protein (CMP) encoding genes (Fig. 1B and Supplementary Table 1) identified in the highly annotated and validated Matrisome 2.0 Database[19,20]. The genes with low standard deviation across samples (s < 1) were filtered out to enable classification of patients with more variant and likely discriminant transcription events leading to a total of 173 genes with high variation and discriminant power across patient samples. In all analyses, hierarchical clustering of the RNA expression levels was performed using Manhattan distance and Ward method (heatmap.2 function in the gplot R package). Survival analysis was performed with the Kaplan–Meier method for censored data (Survival package in R in Fig. 5) and cBioPortal survival analysis module (Figs. 1C–D and 5B) for group comparisons. The statistical significance of survival differences was evaluated with log-rank test. The enrichment analysis of matrisome subtypes across demographic groups and transcriptional subtypes were performed using two-sided Fisher's exact test. In each test, the enrichment of the most common matrisome subtype in a corresponding patient group (e.g., male) was tested against all other matrisome subtypes and other patient groups (e.g., female) using a 2 × 2 contingency table. The statistical difference of Karnofsky performance score and median age across matrisome subtypes were analyzed using a 1-way ANOVA test. The mutation data included whole exome sequencing data from 147 patients for which RNA expression data was also available. The mutational and copy number alteration oncoprints were generated using the oncoprint module in cBioPortal. The mutational and copy number enrichment analyses across matrisome groups were performed based on a two-sided Fisher's exact test followed by Benjamini–Hochberg FDR-correction. The proteomic pathway scores were based on reverse phase proteomics (RPPA) data and the pathway score definitions and gene/protein lists in Akbani et al.[46]. (Supplementary Table 13 in the referenced article[46]). In the referenced paper by Akbani et al.[46], members of each pathway had been predefined based on a Pubmed literature search on review articles describing the various pathways in detail. The batch corrected RPPA data had been median-centered and normalized across all samples to obtain the relative protein level. The pathway score was then the sum of the relative protein level of all positive regulatory components minus that of negative regulatory components in a particular pathway. We extracted the pathway scores for the IDH-wt GBM samples. The pathways included were apoptosis, cell cycle, DNA damage response pathway, core reactive pathway (stroma signature), EMT pathway, hormone receptor pathway, RAS/MAPK, EMT, mTOR/TSC. The transcriptomics based EMT scores were computed using the epithelial and mesenchymal gene signatures as defined by Mak et al.[48]. The vascularization score was calculated using the gene signature defined by Masiero et al.[52].

### Multivariate analysis

For multi-variate analysis, a 156-by-6 data table was created by combining the features of CMP subgroups with two demographic features (age and gender), two disease phenotype features (GBM subtype and *MGMT* Methylation status) and one categorical feature for treatment type using Matlab R2020a Update 5, maci64. GLM was created through the fitglm function of Matlab using the 156-by-6 data table as input, and overall survival (in months) as a response variable. The optional parameter of 'distribution' for the fitglm function was set as 'Poisson' to be consistent with the characteristics of response variable, while default values were used for all other parameters. The 156-by-6 data table has five categorical variables and one continuous variable (age), and fitglm picked a reference category for each of the five variables, i.e., M-H for CMP sub-group, Female for Gender, Classical for GBM Sub-type, Methylated for MGMT Methylation status and Chemoradiation for Treatment. The outputs of the fitglm included Coefficient Estimate (ranged [−1,1]), Standard Error, t-statistic and p-value, and together they quantified the impacts on overall survival when each of the five categorical variables changed from the reference category to other possible values, with positive coefficient and t-statistic indicating increased overall survival than the reference category. Meanwhile, the same panel of outputs quantified the impact of increasing age to overall survival.

### Selection of genes contributing to matrisome subtypes

To select CMPs genes discriminating the matrisome subtypes, we performed multinomial logistic regression with Lasso regularization using the TCGA GBM dataset of mRNA expression. The mRNA expression was quantified by RSEM and followed by log2p1 transformation. In this analysis, 157 GBM samples (44, 58, and 55 GBM samples from M-H, M-La, and M-Lb subtypes, respectively) with 274 GBM core matrisome genes were initially included following exclusion of genes with σ < 1 (#genes = 154). The function cv.glmnet of the R package glmnet was used to do a 10-fold cross validation for selecting an optimal value from a series of regularization parameter λ's. At the optimal λ, where the multinomial deviance is minimal, 47 genes were identified with non-zero coefficients from the multinomial regression model. Finally, from these 47 genes, we obtained 17 genes that were selectively and highly expressed in the M-H subtype and belonged to the same cluster in an unsupervised hierarchical clustering analysis. This 17 gene panel was termed the matrisome 17 (M17) signature.

### Immune cell infiltration and immune checkpoint analysis

The total leukocyte fraction and proportion of immune cell types for the GBM samples in the TCGA cohort were imported from Thorsson et al.[50]. The methylation based total leukocyte fraction had been calculated as explained in Hoadley et al.[49] and captured the leukocyte fraction based on the methylation probes with the greatest differences between pure leukocyte cells and normal tissue followed by estimation of leukocyte content with a mixture model. The proportion of immune cell types with respect to each had been inferred by a CIBERSORT analysis of GBM transcriptomics data[50]. The absolute fractions of immune cell types in each tumor were estimated through multiplication of the proportion of the immune cell types in the immune cell population with total leukocyte fraction to enable cross sample comparisons of specific immune cell types. In total, immune cell infiltration from 153 GBM samples (42 M-H, 58 M-La, 53 M-Lb) were analyzed. The levels of total infiltration and immune cell types were compared between the M-H and M-La/b groups using the Wilcoxon test. For an integrated M-H vs. M-L comparison, the p-values for "M-H vs. M-La" and "M-H vs. M-Lb" were merged with the Stouffer method adjusted for multiple hypothesis testing using the Bonferroni method. For immune checkpoint analysis, we curated 31 therapeutically actionable immune checkpoints, which have been subject of immunotherapy clinical trials[51]. The RNA expression levels of immune checkpoints were analyzed for the same patient cohort as before: 153 samples (42 M-H, 58 M-La, 53 M-Lb). The significance of differences was assessed using a Wilcoxon test followed by a Stouffer meta-analysis to merge the P-values for M-H vs. M-La and M-H vs M-Lb. The resulting p-values were adjusted for multiple hypothesis testing using Bonferroni method.

## IvyGap GBM analyses

Anatomic and histologic domains in which CMPs gene expression was enriched were identified through analysis of the IvyGap GBM database[13]. IvyGap GBM database provides transcriptional signatures from laser capture dissected regions within GBM segmented by histologically defined anatomic structures or enrichment in putative cancer stem cell gene expression identified by RNA in situ hybridization. In this study we included only IDH wild type patients (*n* = 34) and excluded IDH mutant patients. Therefore, the anatomic RNAseq study included 110 samples across 9 patient tumors and cancer stem cell RNA-seq study included 135 samples across 31 patient tumors. The anatomic domains include- Leading edge (LE) at the margin of tumor, Infiltrating tumor (IFT) between leading edge and tumor core; and Cellular tumor (CT) regions comprising the tumor core. Within CT, subregions are identified based on structural features as follows: Hyperplastic Blood Vessels (HBV), Microvascular Proliferation (MVP), Pseudopalisading cells around necrosis (PAN) and Perinecrotic zone (PNZ). Since both reference histology and cancer stem cell sample sets were annotated by the above anatomic domains, they could be combined for regional analysis of correlations with CMPs gene expression. Using the combined RNAseq data we performed unsupervised and supervised hierarchical clustering to identify enrichment of CMPs gene expression and signatures in specific anatomic structures. Gene expression data was aligned against hg19 reference assembly, normalized (TbT normalization method as described by Kadota et al.[88] and analyzed to quantitate FPKM value of each gene. Anatomic and cancer stem cell RNA-seq studies were selected to define structural features and perform hierarchical clustering analysis using MORPHEUS software (https://software.broadinstitute.org/morpheus). RNAseq data from multiple samples within individual patient tumors was used to evaluate intra-tumoral heterogeneity of CMPs gene expression.

## Single Cell RNAseq

Single-cell RNAseq data were mined from GSE182109[33] and processed according to the authors' previously described methods[33]. Data were analyzed using Seurat V4.0.0 (RRID:SCR_016341) and ggplot2 V3.3.3 (RRID:SCR_014601) and ComplexHeatmap V2.7.8.100 (RRID:SCR_017270) packages were used for visualization.

## Proteogenomic analysis

Mass spectroscopy-based proteomics and RNA sequencing data available from the CPTAC glioblastoma repository[53] (https://pdc.cancer.gov/pdc/study/PDC000204) were analyzed to classify patients based on the 17-gene) matrisome gene signature expression. The tissue from 99 patients had been profiled with mass spectrometry analysis using the 11-plexed isobaric tandem mass tags (TMT-11). The RNA expression had been profiled from the matched samples by sequencing on HiSeq 4000 as paired end 75 base pairs[53]. Excluded samples from our analysis were the cases with IDH1 hotspot mutations (6 patients), low-quality cases that failed the pathology evaluation (6 samples with low tumor nuclei, low cellularity, and high necrosis) and one case from a patient who died immediately after surgery due to intracerebral hematoma. The resulting 86 cases carrying IDH1 wild type tumors with high quality based on pathology review are used in the matrisome analysis. Proteomics data covered 13 of the 17 proteins (COL22A1, PODNL1, SNED1, SPON2 proteins were captured in less than half to none of the samples). The *Z*-scores of log-transformed protein expression levels are used for the analysis. The RNA expression data covered all the 17 CMP genes in the signature. RNA expression data is median normalized across the samples and log-transformed. Hierarchical unsupervised clustering analysis (Manhattan distance and Ward's method) was applied to both RNA and protein expression data to select the patients with differential matrisome signature expression. The protein/RNA consensus M17-H cohort which involved the patients with high matrisome signature expression at both RNA and protein level was identified. To define the consensus, first, the patients in the M17-H cluster in the proteomics-based unsupervised analysis were included. Next, to eliminate the cases that are not concordant with RNA expression levels, we quantified a transcriptomic matrisome signature score for each sample as the sum of RNA expression levels across the 17 matrisome signature genes (log normalized values as used in unsupervised clustering). For the M17-H consensus, we filtered out the samples with transcriptional matrisome scores levels below the upper 35% percentile. This filtering provided a group of samples that co-cluster in the high protein expression group and carry high RNA expression that support the observed proteomic levels, meanwhile it eliminated the likely false positives in which the high protein expression is not backed by high RNA levels. We also included the select cases in the M17-I group that carry both high mRNA (>90 percentile) and high proteomic (the median signature protein expression level falls into M17-H cohort range) levels. Similarly, the consensus M17-L cohort was identified to include matrisome low samples in protein levels only when matrisome RNA levels are also below average (lower 50%, Z < 0) to exclude potential false negatives (i.e., low matrisome cases). We included the cases in the M17-I group that carry both low mRNA (<10 percentile) and low proteomic (the median signature protein expression level falls into M17-L cohort range) levels. The overall survival for the M17-H and M17-L levels were compared using Kaplan-Meyer curves and log rank statistics with censored overall survival data.

## Analysis of response to PD1 blockade

The association of CMPs gene expression with immunotherapy was investigated using pre-treatment RNA expression and PD1 blockade (with pembrolizumab) data from predominantly GBM, recurrent grade 4 glioma patients. The mRNA analysis involved 29 patients treated with neoadjuvant or adjuvant anti-PD1 agent pembrolizumab. The RNA expression count data was imported from Gene Expression Omnibus under accession number GSE121810. The clinical data available for 28 patients is based on the reference[34] and clinical trial (NCT02852655) and was kindly provided by Dr. Robert M Prins. The raw count data was subject to counts per million (CPM) normalization and log transformation. Next, the log count normalized expression levels of the genes in the 17-gene matrisome signature were median normalized across the samples and analyzed with a hierarchical clustering (Ward method, Manhattan distance). The progression free and overall survival of patients in the matrisome high vs. low clusters were compared using the Kaplan–Meier curves based on the censored survival data (*N* = 28). The significance of survival differences is assessed with a log-rank test. The median survival times based on Kaplan–Meier curves were reported (survival package in R). The CPM-normalized transcription data, which were from samples of patients after second surgery (recurrence), were analyzed with unsupervised hierarchical clustering (Ward method, Manhattan distance) to identify the M17-H vs. M17-L patients based on the 17-gene matrisome signature. The survival from recurrence to death was approximated through subtraction of the interval between the first (immediately after first diagnosis) and second surgery (at recurrence) from the overall survival. The recurrence to death survival times, log-rank *P* values, and Hazard rations between M17-H vs. M17-L patients were analyzed with Kaplan Meier curves using the Survival package in software suite, *R*.

## Ethics statement

This study was approved by HMRI Institutional Review Board. Informed consent was not required as this study used de-personalized data, collected via routine methods, and shared via public databases. This study was conducted in accordance with the principles of the Declaration of Helsinki.

## Reporting summary

Further information on research design is available in the Nature Portfolio Reporting Summary linked to this article.

## Data availability

All the data sets used in this manuscript are publicly available. The accession codes and links to repositories are provided in methods or other relevant sections. The supplementary data underlying each figure panel is included as a supplementary file.

## Code availability

The pieces of code and scripts which were used to execute the analyses are available upon request.

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

## Acknowledgements
This study was supported by grants John S. "Steve" Dunn, Jr. & Dagmar Dunn Pickens Gipe Chair in Brain Tumor Research Foundation (# 20090011, 2019 SPG Performance grant and NIH funded R01-CA181445. We are also thankful to the Department of Bioinformatics and Computational Biology, at MD Anderson Cancer Center, Houston, TX USA. This work is supported with grants from MDACC Support Grant P30 CA016672 (the Bioinformatics Shared Resource) (A.K.), U01CA253472 (A.K.), UT Austin-MD Anderson Accelarator Grant (AK), NIH R01 5R01NS121405 (KY). This manuscript was edited at Life Science Editors.

## Author contributions
A.K., M.V., Z.D. and R.C.R. conceptualized the study, R.C.R., M.V., M.C.F.C. and A.K. prepared the manuscript, A.K., Z.D., X.L., B.B. (TCGA database analysis), M.V. (IVYGap GBM), Z.Y. (multivariate analysis), M.K., A.K., K.T., and O.B. (proteomics), N.A., E.K.K., A.K. (scRNAseq), K.Y. supervise the scRNAseq analysis, R.C.R. and A.K. supervise the study.

## Competing interests
Kyuson Yun is a co-founder of EMPIRI, Inc. Zeynep Dereli is a co-founder of Vivoz Biolabs.
