## [Transparent Peer Review file · Communications Biology]

A prognostic matrix gene expression signature defines functional glioblastoma phenotypes and niches.

Corresponding Author: Dr Anil Korkut

Version 0:

Reviewer comments:

Reviewer #1

(Remarks to the Author)

Vishnoi M et al used publicly available GBM omics datasets to analyze extra cellular core matrix proteins expression in disease. It is good study used various statistical methods in the analysis. I have following concerns before recommending for publication.

Major concerns

- 1) In methods related to TCGA datasets authors mention they filtered IDH1 mutations it will good if they mention reason for this filter. This information will be useful for readers.
- 2) In result section one authors mention 274 core matrix proteins. It is not clear how this gene set is extracted.
- 3) In result section related core matrix gene expression authors analyse matrix-high and low type and type b. How the low type a and b are defined meaning it is not clear how they arrived at these subgroups.
- 4) In result section related matrixome signature expression GSEA is performed in M-H versus M-L it is not clear if they considered both M-low type and type b. It is also not clear why only RTK pathway is analysed.
- 5) In result section related to matrixome signature as a prognostic markers in GBM. Authors analysed 17 gene-signature of matrixome at transcriptomic in prognostic point of view. They also looked into GBM single-cell RNAseq data but it will be interesting to look into glial cell expression.

Reviewer #2

(Remarks to the Author)

Summary of the manuscript:

The manuscript presents a comprehensive multi-omic study aimed at identifying extracellular matrix (ECM) gene expression signatures associated with glioblastoma (GBM) phenotypes, prognosis, and immune microenvironments. Through the analysis of publicly available datasets (TCGA, IvyGap, CPTAC, and single-cell transcriptomics), the authors identify a 17-gene matrixome signature (M17) that stratifies GBM patients into prognostic groups and potentially predicts responses to PD-1 blockade therapy. The study highlights the role of ECM genes in tumor-stroma interactions, mesenchymal transitions, and immune evasion, providing new insights into GBM biology and clinical stratification.

Strengths of the manuscript:

- Given the ECM plays a prominent role in shaping the tumor microenvironment, the study examines an important question regarding the role of ECM in GBM progression and therapy response. In addition, the M17 signature provides a novel stratification tool that could refine prognostic predictions and guide treatment decisions, and potentially for immunotherapy.
- The integration of transcriptomic, proteomic, spatial, and single-cell data across multiple studies strengthens the conclusions and provides a holistic view of matrixome biology in GBM. The use of multiple independent datasets ensures reproducibility and increases the study's robustness.
- The authors suggest that ECM gene expression may influence immune checkpoint activity and response to anti-PD1 therapy, potentially guiding patient selection for treatment, and the study's overall findings could lead to future biomarker development for clinical applications.
- The authors applied appropriate statistical analyses to support the validity of their findings.

Areas for improvement:

- The introduction and results sections could benefit from improved readability by reducing jargon and simplifying explanations where possible.
 - While the study provides strong correlations between ECM gene expression and GBM phenotypes, there is limited experimental validation. Experimental validation of key matrisome genes using immunohistochemistry (IHC) and functional studies to evaluate the mechanistic role of ECM genes in GBM cells would strengthen the causal relationship between M17 expression and tumor behavior.
 - The study relies heavily on retrospective data. A prospective validation of the M17 signature in an independent GBM cohort would significantly enhance its clinical relevance.
 - Application of a gene signature in a clinical setting may be limited as RNAseq cannot be performed routinely as a clinical test. Developing a clinical assay (e.g., qPCR, IHC panel) or identifying a single gene that can be examined by IHC as a representative of the M17 signature and testing it in patient tumor samples would provide translational value.
 - It would be valuable for the authors to clarify or highlight whether the expression of ICP genes alone vs. the M17 gene signature alone differ in their clinical application to predict response to immunotherapy. It is not clear if M17 provide an advantage or is superior to ICP genes alone for the prediction of response to immunotherapy.
- Final recommendation: The manuscript presents a novel and impactful study on ECM biology in GBM. The findings are compelling, well-supported by data, and have significant translational potential. However, understanding the mechanistic impact of these genes, and establishment of this signature as a clinically applicable tool requires further investigation. I recommend minor revisions before acceptance, with emphasis on validating the clinical utility of the M17 gene signature.

Reviewer's Decision: Minor revisions required.

Version 1:

Reviewer comments:

Reviewer #1

(Remarks to the Author)

authors clarified all my queries I recommend for publication

Authors response to Reviewers' comments

We thank the referees to the reviewers and editor for a constructive review and evaluation of the manuscript. Here, we have addressed all reviewer comments and revised the text as needed. The revisions in the manuscript are highlighted in this letter. Specifically, we have improved the manuscript considering the reviewer comments focusing on clarity of the methods, a stronger discussion on future clinical translation and mechanistic studies based on this work, clarification of analytical decision and inclusion of new analyses to evaluate M17 performance.

Reviewer #1 (Remarks to the Author):

Vishnoi M et al used publicly available GBM omics datasets to analyze extra cellular core matrix proteins expression in disease. It is good study used various statistical methods in the analysis. I have following concerns before recommending for publication.

Major concerns

Review Comment 1) In methods related to TCGA datasets authors mention they filtered IDH1 mutations it will good if they mention reason for this filter. This information will be useful for readers.

Author Response. IDH mutational status is currently a standard factor in classifying malignant glioma due to its validated and robust associations with distinct biological and clinical features of malignancy. IDH wild-type (WT) status is a hallmark of primary glioblastoma (highly aggressive, high-grade, stage 4 glioma) while IDH mutations are now considered as a separate entity [“astrocytoma IDH mutant” and graded 2,3, or 4] due to their distinct demographics (younger, evolving from lower-grade glioma) and significantly better prognosis [WHO 2021]. Therefore, to align with current clinical and diagnostic standards and provide a more defined clinical population for future comparative studies, IDH mutant samples were removed from the analysis. The rationale for filtering TCGA data based on IDH1 mutations is now mentioned in the text. For details, please see Methods section, page 4 under subheading “TCGA datasets-based multi-omic analyses” (Highlighted yellow in color).

Review Comment 2) In result section one authors mention 274 core matrix proteins. It is not clear how this gene set is extracted.

Author response. The matrisome (the entirety of extracellular proteins that comprise the extracellular matrix) is subdivided into Core matrix proteins (CMPs) and Matrix associated proteins (MAPs). The 274 core matrix protein (CMP) genes (Suppl. Table S1) were extracted from a well-validated matrisome database annotated by CMP and MAP designations (PMID: 21937732). The CMPs provide a structural/functional scaffold while MAPs interact with and regulate the Core Matrix to promote cell adhesion and signaling and regulation of ECM composition and structure. The derivation of the CMPs studied here and the rationale is further explained in the revised manuscript For details, please see Methods section, page 4 under subheading “TCGA datasets-based multi-omic analyses” (Highlighted green in color).

Review Comment 3) In result section related core matrix gene expression authors analyse matrix-high and low type and type b. How the low type a and b are defined meaning it is not clear how they arrived at these subgroups.

Author response. These groups were defined based on distributions of CMP gene expression evident in heat maps derived from unsupervised clustering analysis. Matrix Low (ML) group based on the analysis of most variant matrisome genes (N=173 out of 274 genes) exhibited 2 distinct patterns of gene expression distinct from matrix high (Figure 1B, 2b-d and 3). While, the two matrix-low populations were molecularly

different from each other, their mesenchymal, immune and clinical characteristics were similar and differences in overall survival were not significant although we observed a trend for better overall survival in ML-b. The EGFR genomic alterations and pathway activity were higher in ML-b. How the initial unsupervised clustering and survival analyses identified M-H, M-La and M-Lb is described on page 9 and highlighted in green.

Yet, further refinement of the matrisome signature with the final M17 gene signature led to the clinically more meaningful M-H, M-I (intermediate) and M-L cohorts where we observed robust survival and matrisome composition changes (poor survival in M-H, intermediate in M-I and better survival in M-L; Fig 5 A,B, highlighted in cyan on page 1 – last paragraph of introduction.)

Review Comment 4) In result section related matrisome signature expression GSEA is performed in M-H versus M-L it is not clear if they considered both M-low type and type b. It is also not clear why only RTK pathway is analyzed.

Author response. *The associations of M-H, M-La and M-Lb subgroups with patient outcomes indicated a predominant impact of M-H, whereas M-La and M-Lb carried indistinguishable progression survival (Figure 1C and D), mesenchymal (Figure 2D) and immune (Figure 3) characteristics. Based on this observation and for a simplified analysis to better discriminate the clinically most aggressive group (M-H), we focused on a distinction between M-H vs. M-L. To further define clinical, cellular and molecular features which discriminated M-H from non-M-H signatures (i.e., M-La and M-Lb), we combined M-La and M-Lb groups into a single M-L cohort.*

We have analyzed the canonical oncogenic pathways RTK, mTOR, MAPK, AKT, hormone receptor, DNA damage, cell cycle and apoptosis pathways as defined in reference (PMIDs: 24777629 and 31201206). The only significantly altered pathways were RTK and core reactive pathways (PMID: 30297783). The pathway activities with significant changes across matrisome groups are given in figure 2C, the other pathway analyses with no significant difference are provided in supplementary figure 2B. This is also described in methods, page 4 and highlighted in cyan.

Review Comment 5) In result section related to matrisome signature as a prognostic marker in GBM. Authors analyzed 17 gene-signature of matrisome at transcriptomic in prognostic point of view. They also looked into GBM single-cell RNAseq data, but it will be interesting to look into glial cell expression.

We agree that expression in glia is of interest. Glial cells in GBM include both oligodendroglia and astroglial lineage cells. The single-cell analysis demonstrated a distinct CMP gene expression enrichment in oligodendroglia cells. However, a unique astroglial cell cluster was not identified. This is likely due to the known technical difficulties in isolating intact single astrocytes (PMID: 39308429)[<https://doi.org/10.1002/glia.24621>]. Future studies could potentially overcome this challenge through single nuclear RNAseq techniques.

Reviewer #2 (Remarks to the Author):

Summary of the manuscript: The manuscript presents a comprehensive multi-omic study aimed at identifying extracellular matrix (ECM) gene expression signatures associated with glioblastoma (GBM) phenotypes, prognosis, and immune microenvironments. Through the analysis of publicly available datasets (TCGA, IvyGap, CPTAC, and single-cell transcriptomics), the authors identify a 17-gene matrisome signature (M17) that stratifies GBM patients into prognostic groups and potentially predicts responses to PD-1 blockade therapy. The study highlights the role of ECM genes in tumor-stroma interactions, mesenchymal transitions, and immune evasion, providing new insights into GBM biology and clinical stratification.

Strengths of the manuscript:

-Given the ECM plays a prominent role in shaping the tumor microenvironment, the study examines an important question regarding the role of ECM in GBM progression and therapy response. In addition, the M17 signature provides a novel stratification tool that could refine prognostic predictions and guide treatment decisions, and potentially for immunotherapy.

-The integration of transcriptomic, proteomic, spatial, and single-cell data across multiple studies strengthens the conclusions and provides a holistic view of matrisome biology in GBM. The use of multiple independent datasets ensures reproducibility and increases the study's robustness.

-The authors suggest that ECM gene expression may influence immune checkpoint activity and response to anti-PD1 therapy, potentially guiding patient selection for treatment, and the study's overall findings could lead to future biomarker development for clinical applications.

-The authors applied appropriate statistical analyses to support the validity of their findings.

Author response. *We thank the referee for recognizing the key strengths of the manuscript. We have addressed the areas for improvement below.*

Areas for improvement:

- Review Comment: The introduction and results sections could benefit from improved readability by reducing jargon and simplifying explanations where possible.

Author response. *We have revised the introduction and results section with reduced jargon and simplifications to improve readability (Please see Introduction Page 3; Methods Page 4 and Results Page 8-13). (There are many minor changes involving changes in few words, the substantial changes are Highlighted green in color).*

Review Comment 1) While the study provides strong correlations between ECM gene expression and GBM phenotypes, there is limited experimental validation. Experimental validation of key matrisome genes using immunohistochemistry (IHC) and functional studies to evaluate the mechanistic role of ECM genes in GBM cells would strengthen the causal relationship between M17 expression and tumor behavior.

Author response. *We acknowledge the importance of experimental validation, which is the focus of our planned next steps. The primary goal here was to identify core matrisome components in GBM, determine their localization and define prognostic and clinical relevance as a starting point from which to launch subsequent experimental studies to define functional and mechanistic roles of the specific ECM genes and proteins. Experimental approaches required to effectively model regionally defined interplay between core matrix proteins and cell types are extremely complex and include multiplexed spatial IHC analyses among others. The IHC study would require testing a large set of clinically annotated archival or prospective tumor samples while functional mechanistic studies would likely require development of 3D models comprised of multiple cell types and core matrix proteins. While these experiments are expected to shed light on the functional and mechanistic contributions of the identified ECM genes and are important follow-ups to the present study, they are labor intensive and require extensive development and optimization which is beyond the scope of the current manuscript.*

While we have established this study for comprehensive genomic characterization of GBM matrisome, our findings have led to major hypotheses on immune-tumor interactions and enrichment of matrisome genes in distinct GBM niches. Instead of isolated experimental studies that could be partially informative, we plan a comprehensive follow up study including spatial single cell omics studies to further characterize the GBM matrisome in greater detail and likely follow up by functional studies. This will be addressed in a

future study. In the revised version, we discussed these points in the discussion section (See Page 15; Highlighted green in color).

Review Comment 2) The study relies heavily on retrospective data. A prospective validation of the M17 signature in an independent GBM cohort would significantly enhance its clinical relevance.

Author response. *We thank the referee for this highly relevant comment. As a first step in validation of our work based on TCGA data, we validated the value of the M17 signature in the CPTAC proteomics and transcriptomics data (Figure 5 and Supplementary Figure 5). While we heartily acknowledge the value of a prospective study to establish the clinical utility of the M17 panel particularly for treatment response evaluation, this is a major undertaking outside the scope of this manuscript. This would require a pilot feasibility study to validate the assay platform followed by a prospective study of a large number of patients with high quality clinical annotation. To provide statistical power required to validate the M17 prognostic power and its predictive value for responses to specific therapies would require its prospective application in patients enrolled through a multi-center consortium of patients undergoing standardized treatments or as part of a multi-center therapeutic trial. Using our current studies as preliminary, we will seek to establish sponsors and funding for this major but highly necessary clinical study. In the revised version, we have discussed these points in the discussion section (See Discussion Page 15; Highlighted green in color).*

Review Comment 3) Application of a gene signature in a clinical setting may be limited as RNAseq cannot be performed routinely as a clinical test. Developing a clinical assay (e.g., qPCR, IHC panel) or identifying a single gene that can be examined by IHC as a representative of the M17 signature and testing it in patient tumor samples would provide translational value.

Author response. *We agree and appreciate the suggestion. Indeed, this is our long-term goal and the rationale behind this preliminary study. We plan to develop multiplexed qPCR assays and IHC panels to quantify and localize M17 proteins. In addition, we plan to determine whether a smaller matrix signature can provide equivalent prognostic and/or predictive value. These would be the first steps toward establishing the analytic and clinical validity of the M17 or alternative CMP signatures required to formally establish their translational value and ultimately gain approval through the FDA Biomarker Qualification Program. In the revised version, we have discussed these points in the discussion section (See Page 15, highlighted yellow in color).*

Review Comment 4) It would be valuable for the authors to clarify or highlight whether the expression of ICP genes alone vs. the M17 gene signature alone differ in their clinical application to predict response to immunotherapy. It is not clear if M17 provide an advantage or is superior to ICP genes alone for the prediction of response to immunotherapy.

Author response. *We thank the referee for this important comment. We have analyzed the association between patient survival after therapy and the immune checkpoint expression, particularly the immediate targets PD1 as well as PDL1 expression. The ICP expression was not a predictor with statistical significance in this immunotherapy trial. The M17 gene signature is a better predictor of potential therapy response represented by survival after therapy compared to PD1 and PDL1 ICP expression. While we clearly see M17 is superior to ICP expression in this relatively small trial, we think more research is warranted with larger patient cohorts to establish the predictive power of M17 signature compared to ICP or any other marker.*

Here, as we used ICP expression as an important comparison, thanks to the suggestion of the referee. We have also decided to de-escalate the comparison to chemotherapy response to avoid any misleading result. We concluded that a comparison with a completely different cohort is not reliable as varying disease characteristics may confound the results. Also, the major differences in sizes of the two cohorts (anti-PD1 vs. GLASS cohort) prevents a robust comparison between the statistical significance and Hazard Rate

calculations. Therefore, we have chosen to remove this analysis and replace it with the more direct comparison with ICP expression that was performed in the identical cohort and matched patients, a more robust and reliable benchmark for performance of M17-H signature (Please see Figure 5 and revised supplementary Suppl. Figure 5K). The results are also summarized on page 13, highlighted in green and discussed on page 15, highlighted in cyan.

Final recommendation: The manuscript presents a novel and impactful study on ECM biology in GBM. The findings are compelling, well-supported by data, and have significant translational potential. However, understanding the mechanistic impact of these genes, and establishment of this signature as a clinically applicable tool requires further investigation. I recommend minor revisions before acceptance, with emphasis on validating the clinical utility of the M17 gene signature.

Author response. Thank you for the supportive comments. We also agree that there is a need to investigate the mechanistic impact of the genes and determine the utility of the signature as a clinical tool. We are currently making plans for the development of a clinical tool which will be critical to guide additional planned mechanistic studies.